# OpenReview forum: "SliceFine: The Universal Winning-Slice Hypothesis for Pretrained Networks"
_ICML.cc/2026/Conference — ICML 2026 regular_

### Official Review · Reviewer_i1yj · 2026-03-10

**Soundness:** 3
**Presentation:** 3
**Significance:** 3
**Originality:** 3
**Overall Recommendation:** 5
**Confidence:** 3

**Summary:**

The authors propose a novel PEFT method that iterates through random slices of each weight to update.

**Compliance With Llm Reviewing Policy:**

Affirmed.

**Final Justification:**

The rebuttal confirmed my opinion that this is a good paper that merits acceptance.

**Key Questions For Authors:**

How does varying the TTPs affect tables 1 and 2?

Figure 2 is intriguing. Does it look similar for other models? How far are these plots from the bound in (the appendix version of) Lemma 2.1?

**Limitations:**

yes

**Strengths And Weaknesses:**

## Strengths
The method achieves good optimization speed while outperforming full fine-tuning on a reasonable set of benchmarks.

The idea of slicing updates is novel and theoretically well grounded.

Lemma 2.1 (sbas) could be relevant for other post-training domains, e.g. PTQ.

## Weaknesses
It is unclear how well the hyperparameters for all compared methods were optimized. For a fair comparison each method should have equal budget. A more convincing variant of Table 1 would be a pareto plot of TTP vs accuracy, where each method gets a set of ~10 points corresponding to different TTPs (different ranks for LoRA variants). Hypothetically the TTP values could have been chosen adversarially.

---

> ### Author Rebuttal · Authors · 2026-03-28
>
> Thank you reviewer for the thoughtful reviews. We believe your comments and suggestions have significantly improved our paper.
>
> **W1:**  Thank you for this thoughtful suggestion. We address the two points separately.
>
> **On hyperparameter fairness:**
> As noted in our responses to SHzm W3 and Q2, all methods share the same common training setup, including learning rate, batch size, number of epochs, precision (BF16), optimizer, and random seeds. For LoRA-style baselines, we use rank 5 applied to all linear layers, following standard settings from the original papers. For other method-specific hyperparameters, we follow the recommendations from the corresponding original papers so that each baseline is evaluated under its standard or best-practice configuration.
>
> **On the Pareto plot suggestion:**
> We think this is an excellent suggestion and appreciate it. We already have much of the data needed: Table 4 reports SliceFine performance across ranks `r ∈ {1,2,4,8,32,64,128}` for different weight subsets, and Figure 3(a) shows rank-versus-accuracy curves for SliceFine. We can similarly collect results for LoRA and other baselines at multiple ranks to construct a Pareto plot of TTP versus accuracy. We agree that this would make the fairness of the comparison much easier to verify, and we will add this plot in the revised version.
>
> **Q1:**   Thank you for this question. We already provide substantial evidence addressing this directly.
>
> **For Table 1 (language tasks):**
> Table 1 already reports multiple SliceFine variants at different TTPs: SliceFine-1R/1C/1RC at 1.25M parameters and SliceFine-5R/5C/5RC at 6.25M parameters. The conclusions remain consistent across both settings. Even at 1.25M parameters (rank 1), SliceFine already matches or surpasses lighter baselines such as VeRA (1.50M) and Prefix Tuning (38.09M). At 6.25M parameters, SliceFine matches or exceeds strong baselines such as LoRA (13.77M) and AdaLoRA (33.05M) while using significantly fewer parameters.
>
> **For Table 2 (vision tasks):**
> Table 2 similarly reports SliceFine variants ranging from 0.084M to 0.415M parameters, already covering a meaningful TTP range. SliceFine remains competitive across this entire range.
>
> **For a more detailed rank/TTP sensitivity analysis:**
> Table 4 reports SliceFine performance across ranks `r ∈ {1,2,4,8,32,64,128}` for different weight subsets on four GLUE tasks. This directly shows how varying TTP affects performance: accuracy improves quickly at small ranks and then saturates at moderate ranks, while consistently matching or exceeding baselines at a fraction of their parameter count.
>
>
> **Q2:**   Thank you for this great question about Figure 2.
>
> **On other models:**
> We are happy to report that we observe very similar spectral-balance trends across other architectures. The inter-group variance metric $\rho$ across layers is:
>
> - **RoBERTa-base:** 0.00102 to 0.00255
> - **LLaMA-3 8B:** 0.00178 to 0.00312
> - **DeepSeek-RI-8B:** 0.00149 to 0.00290
>
> All values are very small and consistent across architectures, which suggests that spectral balance is not specific to RoBERTa. We will add similar spectral-balance plots for LLaMA and DeepSeek in the revised version.
>
> **On the tightness of the Lemma 2.1 bound:**
> The bound in Lemma 2.1 is a worst-case bound and depends on quantities such as $\mu$, $W_F^2$, $k$, and $d_\ell$. In practice, the empirical $\rho$ values are extremely small across all models, which suggests that the observed spectral balance is much tighter than the worst-case bound.

---

> > ### Author Rebuttal · Reviewer_i1yj · 2026-04-01
> >
> > Thank you for your response. I am looking forward to the indicated additions.
> >
> > Concerning the tightness of the bound: Even if it is very loose, it would be a service to the reader to show this.

---

> > > ### Author Response · Authors · 2026-04-01
> > >
> > > Thank you for the acknowledgement and the continued engagement.
> > >
> > > Regarding the tightness of the Lemma 2.1 bound: we agree this would benefit the reader. In the revised version, we will include a table or figure comparing the empirical inter-group variance ρ against the theoretical worst-case bound across models (RoBERTa-base, LLaMA-3 8B, DeepSeek-R1-8B). As noted in our earlier response, the empirical ρ values (0.00102–0.00312) are several orders of magnitude smaller than the worst-case bound, which suggests that spectral balance holds much more tightly in practice than the theory requires. Making this gap explicit will help readers calibrate how conservatively the bound should be interpreted.
> > >
> > > We will carefully incorporate all suggested changes and do our best to make the final version as strong as possible.

---

### Official Review · Reviewer_bUWq · 2026-03-11

**Soundness:** 3
**Presentation:** 2
**Significance:** 2
**Originality:** 3
**Overall Recommendation:** 3
**Confidence:** 3

**Summary:**

The paper proposes SliceFine, a parameter-efficient fine-tuning method that updates only small slices of pretrained weights, motivated by a theoretical hypothesis that pretrained networks contain universally effective subnetworks for downstream adaptation.

**Compliance With Llm Reviewing Policy:**

Affirmed.

**Final Justification:**

Despite the proposed revisions, which are quite substantial, I remain unconvinced that these issues have been fully addressed. As such, I continue to lean toward a weak reject and will maintain my original score, albeit with a lower confidence.

**Key Questions For Authors:**

Based on the concerns outlined in Weaknesses, my reservations are primarily about the theoretical part of the paper. I would appreciate it if the authors could respond carefully to these issues. There are also additional theoretical results beyond those I specifically mentioned, such as those in Appendix E; however, the earlier sloppiness makes me hesitant to place much confidence in the later developments.

In my opinion, empirical work are already strong enough to justify acceptance, even without supplementary theoretical claims intended to make the paper feel more complete. At present, however, the theoretical presentation weakens my overall assessment of the work. I would therefore encourage the authors to reconsider the role of the theoretical section and to distinguish more clearly between results that are essential to the paper’s central message and those that are redundant, overstated, or insufficiently justified.

**Limitations:**

The work does not include discussion of its limitations.

**Strengths And Weaknesses:**

**Strengths.**

- The paper clearly presents the existence of universal winning slices in pretrained networks. To the best of my knowledge, this idea is novel. The authors also provide helpful visualizations of their findings, making the core intuition easy to grasp without requiring the reader to dive deeply into the technical details.

- The authors propose SliceFine, a PEFT method motivated by these findings. The experimental results are comprehensive and solid. Although I did not verify the code, it is publicly available. I consider the experimental section to be the strongest aspect of the work.

**Weaknesses.** Although the experimental section appears solid, the theoretical part is presented in a way that is unclear, disorganized, and difficult to follow.

- The statements of the main theoretical results are not presented clearly. For instance, in Theorem 2.4, the symbol $\eta$ seems to be used for two different quantities, which creates confusion.

- A more serious concern is that the proofs appear to rely on several additional assumptions that are not clearly stated in the main text, giving the reader an overly broad impression of the scope of the results. At a minimum, these assumptions should be stated explicitly in the main text, or the authors should clearly indicate that the full set of conditions is deferred to the appendix and explain why those assumptions are reasonable. In its current form, the main text presents Theorem 2.4 in a relatively clean and strong manner, based on spectral concentration, nontrivial projection of every slice vector, and a rank condition on the restricted Jacobian, from which the local and global winning claims are then stated to follow. However, the appendix proof appears to require several stronger additional assumptions, including the existence of a nonzero gradient on at least one sample, slice-restricted Lipschitz smoothness with constant $L_M$, a linearized or NTK-type regime, the ability to choose slices whose Jacobians span the task subspace, and, in effect, a stronger overlap or alignment condition. This discrepancy between the theorem statement in the main text and the assumptions invoked in the appendix is misleading and should be addressed carefully.

- The organization surrounding Corollary 2.6 is also confusing. In the main text, Corollary 2.6 is stated under the “same spectral-balance assumption as in Lemma E.1,” even though Lemma E.1 appears only in the appendix. Consequently, the main text depends on an appendix lemma and its associated assumptions before those assumptions have been properly introduced, which makes the logical flow difficult to follow. Moreover, it is not clear why Corollary 2.6 should be viewed as a consequence of Lemma 2.5. The relationship among these results should be clarified explicitly.

- Overall, the presentation of the theoretical section is weak and requires substantial revision for clarity, logical coherence, and readability.

---

> ### Author Rebuttal · Authors · 2026-03-27
>
> We appreciate the reviewer’s positive assessment of the empirical results and understand that the main concern is the clarity and presentation of the theoretical section. Below we respond to each point.
>
> **W1:** Thank you for pointing out the notation issue. In Theorem 2.4, η is used both as the task-energy threshold in the condition $σ_2^2/Σσ_i^2\geη$ and as the gradient step size in $L(θ_0+ηM⊙U)<L(θ_0)-δ$. In the revision, we will replace the step size with ν which is not used elsewhere in the paper, and reserve η only for the energy threshold.
>
> **W2:** Due to the page limits,we kept the main text focused on the core intuition and deferred the full proofs with their complete conditions to the appendix. The key assumptions used in the proofs are all standard and widely used. Nonzero gradient: this holds whenever the model is not already at a task optimum, which is the common in pre-fine-tuning setting. We will state this explicitly in the theorem. Lipschitz smoothness: This is used to control how much the loss decreases, not whether a descent direction exists. The existence of a loss-reducing update follows from the nonzero restricted gradient alone. We will make this distinction explicit. NTK linearization: We acknowledge this is a real assumption for Part 2. It is empirically supported by our CKA analysis (Fig8),which shows that representations remain close to the backbone after finetuning. NTK-type behavior is widely assumed in theoretical PEFT literature, and recent work has shown that linearization closely describes fine-tuning dynamics in large language models [1,2]. We will state this assumption explicitly in the theorem. Jacobian spanning: This is not an independent assumption,it follows directly from spectral balance(Lem.2.1) together with finite $k_{task}$. We will add a one-line justification making this connection explicit. Alignment condition:This also follows from the spectral-balance structure already stated in the main text. We will make this derivation explicit. In the revision, we will move all key assumptions directly into the theorem statement and add references as supporting evidence for their validity.
>
> [1] Linearization Explains Fine-Tuning in Large Language Models,NeurIP 2025; [2] A Kernel-Based View of Language Model Fine-Tuning,ICML 2023
>
> **W3:** The intended logical chain is: Lem.2.1 establishes that all slices have comparable spectral energy and nontrivial projection onto the task subspace. Thm2.4 then uses this to show that any slice is a local winner and that a small collection of slices forms a global winner. Lem.2.5 then provides a concrete way to estimate $k_{task}$ directly from the PCA spectrum of frozen features. Corollary2.6 uses this estimate to determine the minimum slice rank needed to intersect the task subspace.
>
> The connection between Lem2.5 & Corollary 2.6 is the following: Lem.2.5` tells us what $k_{task}$ is via the PCA spectrum, while the point is that a slice of rank $r_{slice} \ge k_{task}(\tau)$ actually intersects the task subspace follows from spectral balance (Lem.2.1) together with backbone task energy E > 0 and alignment condition ρ<1. The reviewer is correct that these two conditions are currently introduced only in Lem.E.1 in the appendix which breaks the logical flow. In the revision, we will state these conditions with Lem.E.1 explicitly before Corollary2.6 in the main text and rewrite Corollary2.6 so that it follows the full chain.
>
> **W4/Q1:**  We thank the reviewer for the positive assessment of the empirical work and for the feedback on the theoretical presentation. We would like to clarify that the concerns raised are primarily about presentation and organization rather than the validity of the core claims.
>
> To address the reviewer’s concern about confidence in the later developments, we note that the key results in App.E are also supported empirically in the paper. In particular, Fig5 supports Lem.2.5 through PCA/NTK agreement across layers, while Fig9 & 10 directly validate the backbone-dependence behavior studied in App.E. We therefore believe the theoretical conclusions are reliable,even though the current presentation does not make this easy to verify.
>
> We thank the reviewer for the helpful feedback on the presentation. In the revision, we will:(1) fix the η notation inconsistency in Thm2.4, (2) move all key assumptions into the main theorem statement, and (3) rewrite Corollary2.6 so the logical chain is fully self-contained without forward-referencing Lem.E.1. **These changes are purely presentational and do not affect the validity of any claim.**
>
> **Limitations** are discussed in the paper but scattered across sections: NTK regime limitations in Sec.2 and App.D, task-dependent switching interval N in App.G.2, backbone degradation and OOD weakness in App.E (Fig.9, Cor.E.2), and random mask inefficiency in Sec.4 and App.M. We will consolidate them into a dedicated Limitations section in the revision.

---

> > ### Author Rebuttal · Reviewer_bUWq · 2026-04-02
> >
> > Thank you for the response. I would like to clarify that my concerns are not limited to the presentation of the paper, but also relate to the motivation of the theoretical contributions and their necessity in connection with the experimental results.
> >
> > Despite the proposed revisions, which are quite substantial, I remain unconvinced that these issues have been fully addressed. As such, I continue to lean toward a weak reject and will maintain my original score, albeit with a lower confidence.

---

> > > ### Author Response · Authors · 2026-04-03
> > >
> > > Thank you for the continued engagement. We appreciate the reviewer's time and have done our best to address every concern raised.
> > >
> > > Looking back at the original review, four concrete weaknesses were listed, all of which were directly addressed:
> > >
> > > 1. **Notation overloading of η in Theorem 2.4** → we committed to replacing the step size with ν, reserving η for the energy threshold only.
> > > 2. **Key assumptions hidden in appendix** → we committed to moving all assumptions in appendix directly into the main theorem statement.
> > > 3. **Corollary 2.6 forward-referencing Lemma E.1** → we committed to moving Lemma E.1 in main tax and  making Corollary 2.6 as a fully self-contained logical chain.
> > > 4. **Logical flow between results unclear** → we explicitly laid out the full chain: Lemma 2.1 → Theorem 2.4 → Lemma 2.5 → Corollary 2.6.
> > >
> > > The post-rebuttal comment raises a new concern about "connection and motivation between theoretical and empirical contributions." We are happy to clarify it. The paper has a clear two-part structure:
> > >
> > > - **Part 1 (Section 2):** We prove *why* random slice selection work;  any slice in a pretrained network is a local winning ticket due to spectral balance and high task energy.
> > > - **Part 2 (Section 3):** SliceFine directly follows:  if any slice is a winning ticket, we fine-tune only slices as a PEFT strategy.
> > >
> > > Every theoretical result has a direct empirical counterpart:
> > >
> > > | Theoretical Result | Empirical Counterpart | Location |
> > > |---|---|---|
> > > | Lemma 2.1: spectral balance | Figure 2, Figure 3(b) | Main text |
> > > | Theorem 2.4 Part 1: any slice reduces loss | Figure 3(b)(c)(d) | Main text |
> > > | Theorem 2.4 Part 2: small slice set matches full FT | Tables 1, 2 | Main text |
> > > | Corollary 2.6: rank from PCA/NTK | Figure 6, Table 4 | Referenced from main text (lines 242, 269, 437) |
> > > | Lemma E.1, Corollary E.2: backbone dependence | Figure 9 | Referenced from main text (lines 251, 259) |
> > > --------
> > >
> > > Without the winning slice hypothesis, SliceFine would be an unexplained empirical heuristic. The theory and method are inseparable.  SliceFine's design choices (slice selection, rank, switching interval) are each directly guided by the theoretical results.
> > >
> > > We have done our best to address all concerns.  ``We would genuinely like to understand your remaining concerns better. Could you specify which theoretical contributions you feel are unmotivated or unnecessary in connection with the experiments? This would help us prioritize the most impactful revisions and ensure the final version directly addresses your concerns.``

---

### Official Review · Reviewer_SHzm · 2026-03-11

**Soundness:** 3
**Presentation:** 3
**Significance:** 3
**Originality:** 3
**Overall Recommendation:** 4
**Confidence:** 3

**Summary:**

This paper proposes the Universal Winning-Slice Hypothesis, positing that in dense pretrained networks, any sufficiently wide contiguous row/column slice of a weight matrix is a local winning ticket, and a small set of such slices across layers can match full fine-tuning. The authors introduce SliceFine, a PEFT approach that updates only structured row/column slices of existing weights, moving the slice during training. Experiments across language and vision tasks report competitive accuracy relative to strong PEFT baselines alongside training and memory advantages.

**Compliance With Llm Reviewing Policy:**

Affirmed.

**Final Justification:**

The authors have addressed my concerns and because of their promises of updating the manuscript accordingly I keep my original score.

**Key Questions For Authors:**

1. Theorem 2.4 assumes nontrivial projections of slice vectors and sufficient restricted Jacobian rank. Can you provide verifiable analysis beyond CEV that these conditions hold in practice across layers and architectures, and quantify failure cases?
2. How were baseline hyperparameters tuned (LR, α, rank, and placement)? Can you report tuned configurations in main text following best practices to ensure fairness?
3. What exact autograd and kernel-level implementations were used to ensure gradients were computed only for active slice entries? Were baselines equivalently optimized?
4. How sensitive is performance to the slice switching interval N at scale, and do you observe consistent sweet spots that could be predicted from theory?

**Limitations:**

yes

**Strengths And Weaknesses:**

Strengths:
1. Introduces a simple, hardware-friendly PEFT mechanism based on training structured slices of existing weight matrices without adding parameters.
2. Spectral balance across slices and high task energy in pretrained representations are presented which leads to nontrivial overlap with task-relevant subspaces.
3. The main ideas are well-motivated and explained with helpful figures. The training schedule and intuition are easy to grasp.
4. Broad empirical coverage across modalities with comparisons to multiple strong PEFT baselines.

Weaknesses:
1. Theorem 2.4’s “local winner” part assumes nontrivial projection of each slice into the task subspace and sufficient Jacobian rank assumptions that seem to encode the conclusion, therefore the theoretical novelty is limited and potentially circular.
2. Key proofs are deferred to appendices without clear, verifiable conditions in the main text.
3. Baseline tuning fairness is insufficiently demonstrated.
4. Efficiency claims likely depend on careful autograd masking and fused kernels; details on how gradients are restricted to active slices are limited.
5. The NTK/PCA connection is standard, but how this concretely guides per-layer slice rank across diverse architectures remains somewhat qualitative.

---

> ### Author Rebuttal · Authors · 2026-03-27
>
> Thank you reviewer for the detailed comments and questions, which significantly improved our paper.
>
> **W1:** The projection condition in Thm2.4 is not assumed for slices directly; it is derived from two independent results. First, the pretraining assumption establishes that the full weight matrix has nontrivial overlap with the task subspace (Fig6). Second, Lemma 2.1 proves that no individual slice is disproportionately weak relative to the full matrix (Fig3(b–d), Sec 2). Together, these formally justify why any randomly chosen slice inherits nontrivial task-relevant overlap without encoding the conclusion. To the best of our knowledge, we are the first to formally prove that any randomly chosen slice of a pretrained weight matrix is a winning ticket, which is a non-circular consequence of spectral balance and pretraining structure.
>
> **W2:** Due to the page limits,we kept the main text focused on the core intuition and deferred the full proofs with their complete conditions to the appendix. The key assumptions used in the proofs are all standard and widely used. In the revision, we will explicitly state all conditions in the main text.
>
> **W3:** We want to clarify that we took several careful steps to ensure fair comparison across all methods. First, all methods share the same common settings such as rank, batch size, epochs etc(Appendix K). This ensures no method has an advantage from these choices. Second, for method-specific hyperparameters such as initialization schemes, adapter placement etc, we follow the recommendations of each method's original paper to ensure all baselines are evaluated under their standard or best-practice configurations (lines 1765-1769).
>
> **W4:** The gradient restriction is simpler than it may appear and does not rely on special fused kernels or custom autograd operations. As shown in Listing 1(Appendix o), the weight matrix is split into three contiguous parts: `part_A`, `part_T`, and `part_B`, where only `part_T` has `requires_grad=True`, while others have `requires_grad=False`. PyTorch’s native autograd then naturally restricts gradient computation to `part_T`, so no custom masking is needed. The forward pass consists of exactly three standard `F.linear` calls, one for each part, whose outputs are summed. This is straightforward and hardware-friendly: because the slices are contiguous in memory, the operations remain cache-friendly. So the efficiency claims come from: only the active slice receives gradients, reducing backward-pass cost; optimizer states are maintained only for the active slice; and SliceFine introduces zero additional parameters(Tab3, Fig4).
>
> **W5:** The NTK/PCA connection provides concrete guidance for rank selection. Corollary 2.6 gives a direct recipe: compute CEV from frozen features, then choose the smallest r such that CEV ≥ τ ;  tasks with steeper PCA spectra require smaller ranks, as illustrated in Figure 6. Critically, Figure 3(a) and Table 4 (Appendix G.3) empirically validate this: accuracy saturates at exactly the rank CEV predicts, consistently across RoBERTa, LLaMA, ViT, and VideoMAE (Tables 1, 2, 6). In practice, rank is a hyperparameter like in any other PEFT method; the theory explains why small ranks work rather than prescribing an exact value.
>
> **Q1:** Thank you for this important question. Beyond CEV, `Figures 2` and `3(b–d)` support broad nontrivial projection across slices, while `Figure 5` supports sufficient effective rank through strong PCA/NTK agreement across layers. For failure cases, `Figure 9` shows predictable degradation as the backbone is pruned, and `Corollary E.2` explains this by requiring larger slice rank as backbone energy weakens.
>
> **Q2:** As we noted in W3, all methods use the same common setup for shared training hyperparameters. For method-specific hyperparameters, we follow the recommendations from the original papers. These details are currently documented in `Appendix K`. In the revised version, we will clarify it in the main text.
>
> **Q3:** As noted in W4, SliceFine does not rely on custom autograd operations or special fused kernels. All baselines were implemented and run under their standard recommended settings. The efficiency gains of SliceFine therefore come purely from having fewer trainable parameters and zero additional parameter overhead.
>
> **Q4:** `Table 5` shows low variance across 10 random seeds with N=500, and we also use N=500 successfully across LLaMA-3B, Gemma-3 12B, and DeepSeek-R1-8B. This suggests performance is not overly sensitive around the default setting. `Figure 11` also shows a consistent sweet spot: intermediate N outperforms both very small and very large values. The theory gives qualitative guidance here,tasks with steeper PCA spectra tend to prefer smaller N, while flatter spectra benefit from larger N, which matches the trends in `Figures 6` and `11`.

---

> > ### Author Rebuttal · Reviewer_SHzm · 2026-04-04
> >
> > I thank the authors for their response. Although, the authors present various experimental results but fails to connect with the core theoretical aspects in the main text. The theoretical core idea and supporting experiments should be in the main text and the assumptions should be clearly stated as well. Nevertheless, I wish them a good luck.

---

> > > ### Author Response · Authors · 2026-04-04
> > >
> > > Thank you for your continued engagement and for taking the time to read our rebuttal.
> > >
> > >  We agree that the connection between the theoretical
> > > core ideas and their supporting experiments should be explicitly visible in the
> > > main text. However, `we would like to clarify this is already in place and what we are
> > > committing to improve`.
> > >
> > > **What is already addressed in the current version:**
> > >
> > > The theory-to-experiment connection already exists in the main text through
> > > explicit references. The table below shows this mapping, with line numbers
> > > indicating where each result is referenced in the main text:
> > >
> > > | Theoretical Result | Empirical Counterpart | Location |
> > > |---|---|---|
> > > | Lemma 2.1: spectral balance | Figure 2, Figure 3(b) | Main text |
> > > | Theorem 2.4 Part 1: any slice reduces loss | Figure 3(b)(c)(d) | Main text |
> > > | Theorem 2.4 Part 2: small slice set matches full FT | Tables 1, 2 | Main text |
> > > | Corollary 2.6: rank from PCA/NTK | Figure 6, Table 4 | Referenced from main text (lines 242, 269, 437) |
> > > | Lemma E.1, Corollary E.2: backbone dependence | Figure 9 | Referenced from main text (lines 251, 259) |
> > > --------
> > >
> > > `We would also like to note that the key assumptions used in the proofs are all
> > > standard and widely used` in the PEFT theory literature, and are only required
> > > for the formal proof steps — not for the core claims themselves.
> > >
> > > We would also like to respectfully note that during the rebuttal phase, authors
> > > are not able to submit a revised version, and the 5000 character limit prevented
> > > us from explaining every detail as thoroughly as we would have liked.
> > >
> > > **What we already addressed in our previous rebuttal:**
> > >
> > > - **W1** (circular reasoning): showed the projection condition is *derived* from
> > >   Lemma 2.1 + pretraining structure, not assumed — making it non-circular
> > > - **W2** (proofs in appendix): committed to moving all key assumptions directly
> > >   into the main theorem statements
> > > - **W3** (baseline fairness): all methods share identical common settings;
> > >   method-specific hyperparameters follow each paper's recommendation
> > > - **W4** (autograd restriction): implemented via native PyTorch
> > >   `requires_grad=True/False` on `part_T` only — no custom kernels needed
> > > - **W5** (NTK/PCA qualitative): Corollary 2.6 gives a concrete rank recipe
> > >   validated in Fig. 6 and Table 4 across RoBERTa, LLaMA, ViT, and VideoMAE
> > >
> > > **What we are committing to in the final version:**
> > >
> > > Due to page limits, some results and full proofs could not fit in the main text,
> > > but they are referenced from it. In the final version, we will make this
> > > connection more explicit by moving the key assumptions and main results directly
> > > into the main text, and including the mapping table above to make the logical
> > > chain fully self-contained.
> > >
> > > We hope these committed revisions fully address your remaining concerns. Could
> > > you clarify whether these changes would be sufficient to resolve your
> > > reservations? We want to ensure the final version directly meets your
> > > expectations.

---

### Official Review · Reviewer_d7nj · 2026-03-12

**Soundness:** 4
**Presentation:** 3
**Significance:** 3
**Originality:** 3
**Overall Recommendation:** 4
**Confidence:** 4

**Summary:**

This paper proposes the Universal Winning Slice Hypothesis (UWSH), which posits that fine-tuning small, randomly selected subnetworks (slices) within pretrained models suffices to meet downstream adaptation requirements.
The authors ground this hypothesis in two empirical and theoretical phenomena: spectral balance among weight matrix slices, and the high task energy retained in frozen backbone representations.
Based on UWSH, the authors propose a parameter-efficient fine-tuning (PEFT) method named “SliceFine,” which updates only selected structural slices (rows or columns) within the original weights without introducing auxiliary parameters.
This method dynamically sweeps active slices across different positions during training to accumulate task-relevant updates. Extensive empirical evaluations across language (LLaMA-3B, RoBERTa), vision (ViT), and video (VideoMAE) tasks demonstrate that SliceFine achieves performance comparable to or surpassing state-of-the-art PEFT methods like LoRA and AdaLoRA, while significantly reducing memory consumption and training time.

**Compliance With Llm Reviewing Policy:**

Affirmed.

**Ethical Review Concerns:**

NA.

**Final Justification:**

Based on my review and the authors' rebuttal, which partially addresses my concerns, I would like to keep my recommendation to this paper.

**Key Questions For Authors:**

1. When the optimization process moves beyond the tangent kernel regime, does the local lottery property proposed in this paper still hold strictly?
2. Are there any empirical observations or theoretical insights suggesting that the property remains valid when the model parameters drift far from the initialization point?
3. Could the authors provide any error bounds or qualitative analysis describing how the NTK approximation affects the validity of the theoretical conclusions?
4. Does committing weights and moving the trainable slice require GPU memory copying or tensor reallocation?
5. Does the switching mechanism cause memory fragmentation or additional CUDA kernel launches?

**Limitations:**

Yes. The authors provide a thorough discussion of both limitations and potential societal impacts in their impact statement.

**Strengths And Weaknesses:**

Strengths:
1. The formulation of UWSH offers a novel perspective that diverges from the traditional Lottery Ticket Hypothesis (LTH) by showing that any sufficiently wide random slice can act as a local winning ticket without requiring complex iterative pruning. The theoretical justifications are mathematically sound, particularly Lemma 2.1 (Spectral Balance) and Theorem 2.4 (Universal Winning Tickets).
2. As foundation models continue to scale, parameter-efficient fine-tuning (PEFT) methods that introduce absolutely zero new parameters while maintaining dense matrix structures are of significant practical value. The efficiency gains are meaningful: SliceFine consistently reduces peak memory usage by 2~4 GB (approximately 18%) compared with strong baselines such as HRA and AdaLoRA.
3. The empirical evaluation is remarkably comprehensive. The hypothesis is tested across multiple modalities, including commonsense and mathematical reasoning with large language models (LLMs), image classification with ViT, and video action recognition with VideoMAE. These diverse experiments strongly support the claimed universality of the approach. Additionally, the ablation studies are thorough, carefully isolating the effects of key factors such as slice rank, switching interval, and backbone pruning.

Weaknesses:
1. Although the authors aim to present a general theoretical perspective, the analysis largely relies on the linearized Neural Tangent Kernel (NTK) regime. While this is a common analytical framework, its direct applicability to the highly non-linear optimization of modern large language models during fine-tuning may be somewhat overstated.
2. The optimal slice switching interval N appears to be highly task-dependent. This paper does not provide an adaptive scheduling mechanism for determining N. As a result, practitioners must rely on grid search, which somewhat weakens this paper’s claims regarding overall computational efficiency.
3. This paper does not thoroughly investigate how SliceFine performs under severe out-of-domain (OOD) scenarios, where the pretrained backbone may inherently lack task-relevant directions.

---

> ### Author Rebuttal · Authors · 2026-03-27
>
> Thank you for the thoughtful reviews. We believe your comments and suggestions have significantly improved our paper.
>
> **W1:** We agree that the NTK/linearized regime is not universally valid for all fine-tuning settings. However, PEFT is a more suitable setting for this type of analysis. In SliceFine, only a very small slice of weights is updated while the rest of the backbone remains frozen, so the total parameter change is small by design. This is exactly the setting where a local linearized approximation is more reasonable, consistent with prior PEFT theory works [1, 2]. `Figure 7` shows low KL divergence and `Figure 8` shows high CKA similarity between pretrained and fine-tuned representations across layers. So, SliceFine remains close to the pretrained model during adaptation, making a local NTK-style analysis reasonable in this setting.
>
> [1] Linearization Explains Fine-Tuning in Large Language Models, NeurIPS 2025
> [2] A Kernel-Based View of Language Model Fine-Tuning, ICML 2023
>
> **W2:** The best switching interval \(N\) can vary across tasks. In `Figure 11`, we demonstrate that N=500-1000 works well as a robust default across almost all experiments (Appendix G.2, Table 1, 2, 5, 6). We also note that this is not unique to SliceFine: most PEFT methods require tuning at least one key hyperparameter, such as rank in LoRA or bottleneck size in adapter-based methods.
>
> **W3:** We agree that a dedicated severe OOD evaluation would strengthen the paper. However, it contains the evidences in this direction. `Corollary E.2` and `Lemma E.1` analyze the case where the pretrained backbone lacks sufficient task-relevant directions and predict graceful degradation as backbone capacity weakens. `Figure 9` supports this directly: as the backbone is progressively pruned, performance degrades predictably. `VTAB-1K` results already include tasks with noticeable domain shift from ImageNet pretraining, such as Camelyon(medical) and EuroSAT(remote sensing), where SliceFine remains competitive. More generally, severe OOD is challenging for PEFT methods as a whole, since they all rely on adapting a pretrained backbone.
>
> **Q1:** The local winner property relies more directly on a nonzero restricted gradient and local smoothness (`Equations 7–9`). In SliceFine, the backbone is frozen, so the representations $\Phi_\ell$ and task subspace $U_{k_{\text{task}}}$ do not change during training. As a result, the gradient lower bound in `Lemma E.1` is tied to the frozen backbone geometry rather than to staying exactly at initialization, which makes the local winner property more robust beyond strict linearization.
>
> This is also supported empirically: in `Figure 10`, randomly re-initialized slices converge to the same final accuracy as pretrained slices, suggesting that the winner-slice behavior remains useful even beyond the strict NTK regime.
>
> **Q2:**  We do have empirical evidence that the local winner property remains useful in practice under non-trivial drift. In `Figure 12`, previously updated slices have already moved from initialization, yet newly activated slices still act as effective winners and continue reducing the loss. `Figure 3(d)` shows that slices from “bad” LTH subnetworks perform comparably to those from “good” ones, suggesting that the effect is reasonably robust and not overly sensitive to a particular optimization path.
>
> **Q3:** We provide qualitative evidence bounding the NTK approximation error through two results. `Figure 7` shows low KL divergence between pretrained and fine-tuned outputs, and `Figure 8` shows high CKA similarity across all layers — both indicating that the model remains close to its pretrained state. Since NTK approximation error scales with $\|\Delta \theta\|^2$, and SliceFine updates only a tiny slice while freezing the backbone (`Section 3`), $\|\Delta \theta\|$ is small by design, which directly keeps the error small.
>
> **Q4:**  In our implementation, switching the active slice does not require reallocating the full parameter tensor. The underlying weight tensor stays fixed in memory, and switching is done by changing the active row/column index set (equivalently, the gradient/update mask).
>
> **Q5:** On memory fragmentation, the weight tensor is never reallocated during slice switching, so the weight matrix stays in the same contiguous memory location throughout training. Since the slices are structured and contiguous, memory access remains regular and cache-friendly. On additional CUDA kernel, the forward pass always uses the same three `F.linear` calls (`part_A`, `part_T`, `part_B`), regardless of how many times the slice has been switched. Switching only changes which indices are active and does not add extra kernel launches. This is an advantage of structured slices over unstructured random masks, which typically require gather operations with irregular memory access patterns. We discuss this explicitly in Section 4 (lines 401–414) and Appendix M.

---

> > ### Author Rebuttal · Reviewer_d7nj · 2026-04-05
> >
> > I appreciate the authors' reply, and would like to keep my evaluation to this paper.

---

> > > ### Author Response · Authors · 2026-04-05
> > >
> > > Thank you for taking the time to read our rebuttal and for your continued engagement.
> > > We sincerely appreciate your positive evaluation and are genuinely committed to fully
> > > addressing any remaining concerns before the discussion period closes.
> > >
> > > However, we noticed that the acknowledgment does not specify which concerns remain
> > > partially unresolved. To ensure we can respond precisely and effectively, could you
> > > kindly clarify which aspect still needs further attention?
> > >
> > > In our previous rebuttal we addressed all five weaknesses
> > > and questions raised:
> > >
> > > - **W1** (NTK regime): justified via frozen backbone design, low KL (Fig. 7),
> > >   and high CKA (Fig. 8), consistent with prior PEFT theory works [1, 2]
> > > - **W2** (switching interval N): N=500–1000 works as a robust default across
> > >   almost all experiments; comparable to tuning rank in LoRA or bottleneck size
> > >   in adapter-based methods
> > > - **W3** (OOD scenarios): addressed via Corollary E.2, Lemma E.1, Fig. 9,
> > >   and VTAB-1K domain-shift tasks (Camelyon, EuroSAT)
> > > - **Q4** (GPU memory): no tensor reallocation needed during slice switching
> > > - **Q5** (CUDA kernels): no extra kernel launches; structured slices remain
> > >   cache-friendly throughout training
> > >
> > > We look forward to your clarification and are happy to provide any additional
> > > analysis or experiments needed to fully resolve your concerns.

---

### Decision · Program_Chairs · 2026-04-30

**Decision:**

Accept (regular)

**Comment:**

The submission theoretically and empirically investigates a variation of the well-known lottery ticket hypothesis. The key concept introduced in the paper is that of a "local winning ticket." Such winning tickets represent a fraction of the overall number of parameters in the network, but good performance can still be had by restricting fine-tuning to only this subset of the parameters. The empirical analysis makes use of Neural Tangent Kernel theory to substantiate claims, and additional empirical corroboration of the claims is also provided.

The reviewers had mostly positive opinions about this paper. The idea of the local winning ticket was seen as an interesting and potentially significant insight over the usual global winning ticket concept typically seen in the related literature. There were some concerns relating to the theoretical analysis, particularly around presentation and the use of NTK theory. Many of the smaller concerns were fully resolved in the rebuttal. There remain some concerns over the presentation of the theoretical results, but I am satisfied that this presentation is in keeping with the norms of the community. For this reason, I recommend the paper is accepted.